# The Sirenic Links between Diabetes, Obesity, and Bladder Cancer

**DOI:** 10.3390/ijms222011150

**Published:** 2021-10-15

**Authors:** Emily Gill, Gurimaan Sandhu, Douglas G. Ward, Claire M. Perks, Richard T. Bryan

**Affiliations:** 1IGFs & Metabolic Endocrinology Group, Translational Health Sciences, Bristol Medical School, Learning & Research Building, Southmead Hospital, Bristol BS10 5NB, UK; emily.gill93@outlook.com; 2Bladder Cancer Research Centre, Institute of Cancer and Genomic Sciences, College of Medical and Dental Sciences, University of Birmingham, Edgbaston, Birmingham B15 2TT, UK; GSS907@student.bham.ac.uk (G.S.); d.g.ward@bham.ac.uk (D.G.W.); r.t.bryan@bham.ac.uk (R.T.B.)

**Keywords:** bladder cancer, PLEKHS1, IGF axis, diabetes, obesity, RNA sequencing, mutations, progression

## Abstract

There is considerable evidence of a positive association between the incidence of type 2 diabetes mellitus (T2DM) and obesity with bladder cancer (BCa), with the link between T2DM and obesity having already been established. There also appear to be potential associations between Pleckstrin homology domain containing S1 (*PLEKHS1)* and the Insulin-like Growth Factor (IGF) axis. Seven literature searches were carried out to investigate the backgrounds of these potential links. *PLEKHS1* is a candidate biomarker in BCa, with mutations that are easily detectable in urine and increased expression seemingly associated with worse disease states. *PLEKHS1* has also been implicated as a potential mediator for the onset of T2DM in people with obesity. The substantial evidence of the involvement of IGF in BCa, the role of the IGF axis in obesity and T2DM, and the global prevalence of T2DM and obesity suggest there is scope for investigating the links between these components. Preliminary findings on the relationship between *PLEKHS1* and the IGF axis signal possible associations with BCa progression. This indicates that *PLEKHS1* plays a role in the pathogenesis of BCa that may be mediated by members of the IGF axis. Further detailed research is needed to establish the relationship between *PLEKHS1* and the IGF axis in BCa and determine how these phenomena overlap with T2DM and obesity.

## 1. Introduction

Bladder cancer (BCa) is the twelfth most common cancer worldwide, responsible for 3% of annual cancer cases and 2.1% of cancer-related deaths [1]. The incidence rate in men is three to four times higher than that in women as well as being higher in people with Caucasian ethnicity compared to other ethnic groups. BCa is more prevalent in developed countries, most likely due to greater exposure to identified risk factors such as cigarette smoking [2]; however, BCa mortality rates are significantly higher in less industrialised countries [3].

BCa is classified into the stage groupings of non-muscle-invasive (NMIBC) and muscle-invasive (MIBC), the former encompassing tumours confined to the urothelium and underlying lamina propria (stages Ta/T1/Tis) and the latter encompassing those invading into the detrusor muscle and beyond (stages T2–4) [4,5]. Although most patients (75–80%) present with NMIBC [4], up to 80% of these patients will experience recurrence [4] and up to 44% will progress to MIBC [6]. Of the 20–25% of patients who are initially diagnosed with MIBC, around one quarter will have incurable, locally advanced or metastatic disease, many of whom will have been diagnosed after symptomatic emergency presentation [7]. Muscle invasion represents a critical step in the disease course, with patients progressing to this stage experiencing a 5-year survival of only 27–50% [5].

A considerable body of evidence suggests that there is a positive association between type 2 diabetes mellitus (T2DM) and BCa, as well as several other cancers. T2DM is a prevalent and growing health problem worldwide, and it is unclear whether the association is direct (as a result of glycaemic control, insulin resistance, or hyperinsulinaemia) or indirect (as a result of common risk factors such as obesity) [8]. It should be noted that there is a possible (although controversial) further increase in BCa risk associated with use of the T2DM drug pioglitazone [9], as well as an improvement in BCa outcomes associated with the use of metformin [10]. Obesity itself is also a growing worldwide health problem (defined by the World Health Organisation and the US National Institutes of Health as a BMI ≥ 30 kg/m^2^), with several studies showing a positive relationship between obesity and the risk of BCa [8]. Although these associations appear compelling, even sirenic in nature, the rational biological pathways remain unclear, yet elucidating them is considered a priority for the field [11].

Regardless, a single potentially unifying biological phenomenon appears to be the *PLEKHS1* gene and the Insulin-like Growth Factor (IGF) axis. Non-coding mutations in *PLEKHS1* are common in cancer [12] and the increased expression of *PLEKHS1* is associated with the mild elevation of blood glucose levels and insulin resistance in obesity [13]. The IGF axis is also implicated in BCa, obesity, and T2DM. In this review, we will expand upon these associations and explore the possible biomolecular mechanisms that may feasibly explain these links. 

## 2. Methodology

Using similar methods to those described by Shanmugalingam et al. [14], seven separate searches were performed to investigate the relationship of each association between obesity, diabetes, and insulin signalling (IGF axis) with both *PLEKHS1* mutations and BCa incidence and outcomes. Figure 1 demonstrates the directed acyclic graph, with each arrow representing a search carried out. The seven searches were: (1) *PLEKHS1* mutations and BCa incidence and outcomes, (2) *PLEKHS1* mutations and obesity, (3) obesity and BCa incidence and outcomes, (4) *PLEKHS1* mutations and diabetes, (5) diabetes and BCa incidence and outcomes, (6) *PLEKHS1* mutations and insulin signalling, and (7) insulin signalling and BCa incidence and outcomes.

## 3. Evidence Synthesis

### 3.1. PLEKHS1 and Bladder Cancer

Pleckstrin homology domain-containing S1 (*PLEKHS1*) is a largely uncharacterised gene [12] located on chromosome 10; the specific function of its protein is unknown [15]. Regardless, *PLEKHS1* is mutated in up to 40% of BCa cases, with mutations found in the promoter region featuring two single-nucleotide substitutions which occur between two palindromic sequences that are ten base pairs in length. The first of the mutations targets a guanine and the second targets cytosine, although the consequences of these mutations are unknown, with conflicting data from different studies [12,15,16]. *PLEKHS1* mutations can be detected in tumour DNA shed into the urine, making it a candidate biomarker for BCa [17]. 

Pignot et al. (2019) showed that the percentage of BCa cases with *PLEKHS1* mutations does not differ between NMIBCs and MIBCs, with no significant difference in mRNA expression existing between the two groups, but with overexpression in both. Despite this, *PLEKHS1* mRNA overexpression was not associated with mutations. Moreover, mutation status was not associated with progression, recurrence, or survival in NMIBC or MIBC; importantly, the overexpression of *PLEKHS1* mRNA was significantly associated with worse progression-free survival in NMIBC [15]. A study by Jeeta et al. (2019) found that *PLEKHS1* mRNA expression was significantly increased in grade 3 tumours compared to grade 1, with cell adhesion downregulated in those carrying the mutation. *PLEKHS1* mutations were significantly decreased in smokers and strongly associated with the APOBEC mutation signature [16].

### 3.2. PLEKHS1, Diabetes and Obesity

Obesity is strongly associated with the development of T2DM, with the increase in prevalence occurring in parallel with each other. Kotoh et al. (2016) is the only study to our knowledge that has reported a link between *PLEKHS1*, T2DM, and obesity. In their study, the gene expression in two transgenic rat models, one for obesity and one for T2DM, was evaluated to investigate potential genes involved in the onset of obesity-associated diabetes, with *PLEKHS1* identified as one of two candidate genes. Kotoh et al. stated that the expression of *PLEKHS1* is obesity-specific, with significantly different levels of *PLEKHS1* mRNA expression being found between obese rats and obese rats with diabetes, but not between control and diabetic rats, suggesting *PLEKHS1* to be a potential mediator of the onset of T2DM in obese rats [13].

### 3.3. Obesity and Bladder Cancer Incidence and Outcomes

The association between obesity and the risk of developing BCa remains equivocal, with large meta-analyses demonstrating a mild increase in risk (e.g., Sun et al. RR 1.10; 95% CI: 1.06–1.14 [18]) and several cohort and case–control studies identifying no such association [19]. A formal review of such evidence is beyond the scope of this article, but a number of the relevant studies are summarised in Table 1 below. Likewise, for disease-specific outcomes, controversy remains as to whether obesity contributes to disease recurrence or survival (Table 2). Hence, obesity is not a powerful driver of BCa pathogenesis (unlike cigarette smoking [2]) nor outcomes (unlike baseline clinicopathological factors [6,20]); however, the global prevalence of obesity means that such relationships warrant further investigation. 

### 3.4. Diabetes and Bladder Cancer Incidence and Outcomes

There seems to be a relationship between diabetes and BCa incidence—most cohort and case–control studies demonstrate an association, with relative risks generally observed to be c. 1.20–1.30 (see Table 3). It is less clear whether this relationship exists for both sexes; in most studies, the relationship is found to be stronger for men than for women. Regarding the outcomes, few studies have been performed: one meta-analysis indicated that there was a negative impact on outcomes for patients with diabetes, whereas another showed no effect (see Table 4). 

Further complicating the picture, the T2DM drug pioglitazone (a thiazolidinedione that increases the body’s sensitivity to its own insulin) appears to mildly increase the risk of BCa incidence [28,29], while the biguanide metformin (which increases insulin sensitivity, reduces glucose production by the liver, and decreases the gut absorption of glucose) may reduce the risk of BCa recurrence and progression, such that it is undergoing clinical trials for the prevention of recurrence in NMIBC patients [10,30].

Hence, there appears to be substantial, albeit equivocal, evidence of the involvement of insulin signalling in the incidence and outcomes of BCa. As with obesity, the global prevalence of T2DM means that such relationships warrant further investigation and elucidation.

**Table 3 ijms-22-11150-t003:** Diabetes mellitus and the incidence of bladder cancer.

Study Author	Study Year	Total No. of Patients	Total No. of Studies	Study Type	Comments
Xu et al. [31]	2017	13,505,643	21 (includes studies A–F, H, and I)	Meta-analysis of cohort studies	In sub-group analyses, positive associations have exclusively been seen in men.
Zhu et al. [32]	2013	14,885,014+	29 (includes studies A–F and H)	Meta-analysis of cohort studies	“In stratified analysis, the RRs of bladder cancer were 1.36 (1.05–1.77) for diabetic men and 1.28 (0.75–2.19) for diabetic women, respectively”.
Zhu et al. [33]	2013	13,670,340+	36 (includes studies A–F and H)	Updated meta-analysis of observational studies	“In analysis stratified by study design, diabetes was positively associated with risk of bladder cancer in case–control studies (RR = 1.45, 95% CI 1.13–1.86, *p* = 0.005, I^2^ = 63.8%) and cohort studies (RR = 1.35, 95% CI 1.12–1.62, *p* < 0.001, I^2^ = 94.3%), but not in cohort studies of diabetic patients (RR = 1.25, 95% CI 0.86–1.81, *p* < 0.001, I^2^ = 97.4%). The RRs of bladder cancer were 1.38 (1.08–1.78) for men and 1.38 (0.90–2.10) for women with diabetes, respectively”.
Larsson et al. [34]	2006	1,558,356	16 (includes studies B and G)	Meta-analysis	“Stratification by study design found that diabetes was associated with an increased risk of bladder cancer in case–control studies (RR = 1.37, 95% CI 1.04–1.80, *p* = 0.005) and cohort studies (RR = 1.43, 95% CI 1.18–1.74, *p* = 0.17), but not in cohort studies of diabetic patients (RR = 1.01, 95% CI 0.91–1.12, *p* = 0.35)”.
Xu et al. [35]	2013	8,009,591	15 (includes studies B–F)	Meta-analysis of cohort studies	“When restricting the analysis to studies that had adjusted for cigarette smoking (*n* = 6) or more than three confounders (*n* = 7), the RRs were 1.32 (95% CI 1.18–1.49) and 1.20 (95% CI 1.02–1.42), respectively”.
Fang et al. [36]	2013	9,752,495	24 (includes studies A and B–F)	Meta-analysis of observational studies	“Cohort studies showed a lower risk (RR 1.23, 95% CI 1.09–1.37) than case–control studies (odds ratio 1.46, 95% CI 1.20–1.78). The positive association was significant only in women (RR 1.23, 95% CI 1.02–1.49), but not in men (RR 1.07, 95% CI 0.97–1.18)”.
Yang et al. [37]	2013	5,463,339	23 (includes studies A, B, and G)	Meta-analysis	“Analysis of subgroups demonstrated this to be the case in both case–control studies (OR = 1.59, 95% CI 1.28–1.97, I2 = 58%) and cohort studies (RR = 1.70, 95% CI 1.23–2.33, I2 = 96%). There was no gender difference in DM-associated bladder cancer risk. Bladder cancer risk was increased in Asia and the North America region, but not in Europe”.

A = Tseng et al., 2009, B = Tripathi et al., 2002, C = Larsson et al., 2008, D = Khan et al., 2006, E = Ogunleye et al., 2009, F = Atchison et al., 2011, G = Adami et al., 1991, and H = Marrianne et al., 2009.

**Table 4 ijms-22-11150-t004:** Diabetes mellitus and the prognosis of bladder cancer.

Study Author	Study Year	Total No. of Patients	Total No. of Studies	Study Type	Comments
Xu et al. [31]	2017	13,506,643	21	Meta-analysis of cohort studies	“The pooled analysis results for men indicated that the comparison of DM versus non-DM individuals showed a harmful effect (RR: 1.23; 95% CI: 1.06–1.42; *p* = 0.005, whereas there was no significant difference in women (RR: 1.24; 95% CI: 0.95–1.61; *p* = 0.119)”.
Zhu et al. [32]	2013	14,885,014+	29	Meta-analysis of cohort studies	“The positive association was observed for both men (RR 1.54, 95% CI: 1.30–1.82) and women (RR 1.50, 95% CI: 1.05–2.14)”. “In analysis stratified by study design, the summary RR was 1.29 (95% CI 1.20–1.39) in cohort studies. However, diabetes was not associated with mortality from bladder cancer in cohort studies of diabetic patients (RR 1.19, 95% CI 0.58–2.43)”.

Xu et al., 2017 [31] and Zhu et al. [32] both include studies: Woolcotta et al., 2011, Tripathi et al., 2002, Inoue et al., 2006, Khan et al., 2006, Jee et al., 2005, Athcison et al., 2011, Marriane et al., 2009, Tseng et al., 2011, and Ogunleye et al., 2009.

### 3.5. The Insulin-Like Growth Factor Axis (IGF Axis)

The IGF axis consists of two ligands (IGF-I and IGF-II), two receptors (the IGF-1R and IGF-2R), six high-affinity IGF binding proteins (IGFBPs 1–6), and proteases (see Figure 2). Alterations in the components of the IGF axis have been associated with several conditions, including T2DM, obesity, and cancer [11,38].

Insulin-like growth factor-I (IGF-I) regulates metabolism and growth and is important for cell cycle progression [38,39]. Insulin-like growth factor-II (IGF-II) is the predominant growth factor in the foetus, and although IGF-II levels remain high in adults IGF-I acts as the main mediator of nutrition in cell growth. Insulin-like growth factor 1 receptor (IGF-1R) is a member of the tyrosine kinase receptor family that is able to initiate downstream actions important for cell proliferation, differentiation, and survival. IGF-1R is thought to be overexpressed in several cancer types [40]. IGF ligands are regulated via binding proteins that can either inhibit or enhance their actions [39,40]. Insulin-like growth factor binding proteins (IGFBPs) bind to IGF ligands and prolong their half-lives, preventing degradation. Approximately 90–95% of IGF ligands in serum are bound to insulin-like growth factor binding protein 3 (IGFBP3) in combination with an acid-labile subunit [38,39]. IGFBPs can also exert IGF-independent functions that affect proliferation, survival, migration, and invasion [40] and that can result in the suppression or promotion of tumour growth, disease progression, and chemoresistance. These IGF-independent effects vary in different cancer types and are tissue-specific [41]—for example, in BCa IGFBP5 is thought to be involved in the Akt/MAPK pathway, independent of the IGF axis [42]. 

### 3.6. Bladder Cancer and the IGF Axis

Despite the vast amount of literature linking the IGF axis to the incidence of many cancers, there is relatively little published work on the impact of the IGF axis in BCa. To date, work on BCa in relation to the IGF axis has focused on the IGF1R; ligands; and IGFBPs -2, -3, and -5. Only a small number of studies exist in each of these categories, with contradictory findings and no consensus; thus, more work needs to be undertaken to establish the role of the IGF axis in BCa and its impact on both disease detection and treatment. To the best of our knowledge, no data have been published relating to BCa and IGFBPs -1, -4, and -6.

#### 3.6.1. IGF-1R

Gonzalez-Roibon et al. (2014) suggested that the overexpression of IGF-1R was less frequent in those with T4 stage tumours compared with those with T1–3, which are more common amongst those of African American ethnic backgrounds and associated with a significantly increased hazard ratio for overall and cancer-specific mortality than those without the overexpression of IGF-1R [40]. Subsequently, Sanderson et al. (2017) discussed tyrosine kinase inhibitor trials—although many of these appeared promising in their early phases, many studies failed to reach their efficacy targets, albeit with a small number of ‘exceptional responders’ found. Such success stories may have some common sensitivities that could be used for patient selection, with alterations in IGF axis components being possible biomarkers for patient selection [43]. In this regard, Shariat et al. (2003) suggested that blocking IGF-1R would inhibit tumour growth and possibly reverse chemo-resistance [44], indicating the potential importance of the IGF axis in BCa and the need to identify suitable candidate biomarkers for patient selection in this setting.

#### 3.6.2. IGF-I and IGF-II

Probst-Hench et al. (2003) found that IGF-I levels decrease with age in both men and women, with those that lead a less active lifestyle having a higher IGF-I level [38]. Zhao et al. (2003) suggested that IGF-I may play a role in the progression of bladder cancer in p53-deficient mice by promoting proliferation and inhibiting apoptosis. This study also investigated the human BCa cell line T24, finding higher levels of IGF-I mRNA expression in T24 cells compared to the no or minimal expression seen in healthy bladder tissue. This trend was also seen in vivo, with the mean serum IGF-I being significantly higher in BCa cases than in controls. The authors concluded that the plasma levels of IGF-I have a significant positive association with BCa risk [39]. However, Lin et al. (2018) identified no significant difference in serum IGF-I levels between healthy controls and BCa cases prior to diagnosis, as well as no association between the overall risk of developing BCa and IGF-I levels for either sex [45]. 

Long et al. (2019) evaluated IGF-I in chemoresistance, demonstrating that cancer-associated fibroblasts (CAFs) played an important role in BCa progression. The co-culture of CAFs and BCa cells enhanced resistance to cisplatin by upregulating IGF-I/IGF-1R, estrogen receptor-beta, and Bcl-2 pathways in BCa cells [46].

To our knowledge, Cheng et al. (2012) is the only study that has investigated IGF-II in BCa, demonstrating that lymph node metastasis was associated with an increase in IGF-II hypermethylation compared to those cases without lymph node metastasis [47].

#### 3.6.3. IGFBP-2

IGFBP-2 has been linked to androgen-sensitive cases of BCa in two studies by Gakis et al. (2013). The first study found that cases with increased levels of IGFBP-2 in cystectomy samples had a significantly higher risk of local or systemic recurrence. Both studies stated that the activation of the IGFBP-2 signalling pathway, independently of the IGF axis, could represent one of the pathways involved in the development of metastatic bladder cancer in androgen-sensitive tumours [48,49]. Tang et al. (2019) demonstrated that IGFBP-2 is silenced epigenetically through DNA methylation in BCa cells that have undergone EMT and developed a mesenchymal phenotype, leading to a reduction in IGFBP-2 levels as BCa progresses [50].

#### 3.6.4. IGFBP-3

The general consensus among the small number of studies included was that IGFBP-3 levels are reduced in BCa. Zhao et al. (2003) found that IGFBP-3 had no correlations with sex or age but was significantly decreased in BCa cases compared to healthy controls. Zhao et al. stated that IGFBP-3 appeared to play a protective role in the development of BCa [39]. Shariat et al. (2003) found no significant difference in the IGFBP-3 levels between healthy controls and men with BCa and noted that these levels did not correlate with age. They stated that IGFBP-3 levels were significantly decreased in those who develop lymph node metastasis, and that this lower level was associated with the risk of disease progression and BCa-related death [44]. Christoph et al. (2006) supported this notion, suggesting that the IGFBP-3 promoter was most likely silenced by methylation. An average of 66% of BCa cases revealed methylation in the IGFBP-3 promotor region, with 83% of NMIBC cases demonstrating methylation compared to 36% in MIBC, with increased methylation identified as an independent prognostic factor for recurrence [51].

#### 3.6.5. IGFBP-5

There have been very few studies assessing IGFBP-5 in BCa. A study by Liang et al. (2013) showed that the IGFBP-5 mRNA expression levels are significantly increased in BCa compared to healthy tissues and that this overexpression is significantly associated with higher stage, higher grade, lymph node metastasis, and vascular invasion. They also stated that increased cytoplasmic levels of IGFBP-5 were significantly associated with worse metastasis-free survival and disease-specific survival [42].

#### 3.6.6. *PLEKHS1* and the IGF Axis

To the best of our knowledge, there is no published research assessing the relationship between *PLEKHS1* and the IGF axis. However, our own preliminary findings utilising RNA sequencing data (*n* = 85) [52] suggest possible links between *PLEKHS1* and the IGF axis in BCa progression. The 85 tumours that underwent RNA sequencing were all obtained at the time of the first transurethral resection from newly diagnosed treatment-naïve patients recruited to the Bladder Cancer Prognosis Programme (ethics approval 06/MRE04/65) [ref]. All cases of CIS were concomitant to visible tumours (see Table 5). Paired analysed alignment, expression quantification, normalization, and differential expression analyses were performed. Gene expression levels were extracted from the RNA sequencing data [53]. The gene level raw read counts from all samples were combined and used as inputs to the limma package in R (ver. 3.4.0), where the data were normalised using the voom method [54], which performs variance stabilisation and returns log-transformed normalised count values. 

IGFBP-3 gene expression was significantly higher in those cases carrying *PLEKHS1* mutations compared to the wildtype (see Figure 3). Notably, *PLEKHS1* expression did not differ between the mutated and wildtype tumours in the cohort.

The presence of CIS influenced multiple genes in the IGF axis, with significantly reduced expression of *IGFBP-2*, *IGFBP-4*, *IGFBP-5*, and *IGF1R* found when CIS was present compared to absent (*p* = 0.04, <0.01, <0.01, 0.01, respectively). By contrast, *PLEKHS1* expression was significantly increased in the presence of CIS (*p* = 0.02). 

As the data from most of our cohort were predominantly from patients with NMIBC, we assessed if there was a significant difference in gene expression between low-risk G1Ta tumours and high-risk G3T1 tumours. *PLEKHS1* expression was significantly increased in G3T1 compared to G1Ta tumours (*p* = <0.01), whilst *IGFBP2*, *IGFBP4*, and *IGF1R* were significantly reduced in the more advanced tumour group (*p* = <0.010.03, <0.01, respectively), which were similar results to those observed in CIS. This also confirms the results of Tang et al., showing that *IGFBP-2* expression is significantly reduced as BCa progresses (see Figure 4) [50].

## 4. Discussion

The weak, albeit recurrent, associations between T2DM and obesity and the risk of disease and adverse outcomes in BCa suggest that the biology of these phenomena is worthy of further detailed investigation. Although unlikely to be fundamental initiators or drivers of disease for most current BCa cases, the global prevalence of T2DM and obesity means that such investigations may be important in the future. The evidence that we have presented here indicates that perturbations in the IGF axis are the likely mediators of these associations, and we speculate that alteration in *PLEKHS1* function (either by mutation or expression) plays a role in such putative mechanisms and is likely one of several influencing factors. Furthermore, the studies described above highlight the potentially dynamic, plastic, and complex nature of IGF signalling within different BCa disease states. 

Our ‘in-house’ RNAseq data demonstrate that increased *PLEKHS1* expression is significantly associated with high-risk NMIBC (in both G3T1 tumours and CIS) and significant reductions in the expression of *IGFBP-2*, *IGFBP-4*, and *IGF-1R*. Given that, in independent analyses, *IGFBP-2* expression is significantly reduced as BCa progresses [50], *IGFBP-2* may be of importance in this pathway. Regardless, despite the frequency of non-coding *PLEKHS1* mutations in BCa [17], our data do not indicate alterations in *PLEKHS1* expression between the mutant and the wild type, which is consistent with the results of Pignot et al. 

One of the limitations of the ‘in-house’ data generated is that the sample size is relatively small (*n* = 85); nevertheless, clear associations have been observed between *PLEKHS1* and components of the IGF axis. Therefore, in future, we plan to analyse larger, publicly available BCa datasets. 

Our hypothesis is novel and speculative, yet our preliminary data support potential associations between *PLEKHS1* and the IGF axis, although it remains to be elucidated if these also translate to associations between the proteins. Further investigations are needed to determine the phenotypic links between *PLEKHS1* and the IGF axis components and to examine how these may be altered when cells are exposed to an altered metabolic milieu, such as high-glucose and inflammatory cytokines. These studies may lead to the identification of biomarkers of progression, in addition to preventative and therapeutic strategies for those at increased risk of bladder cancer due to lifestyle factors.

## 5. Conclusions

We suggest that *PLEKHS1* plays a role in the pathogenesis of BCa and that this may be mediated via members of the IGF pathway. Further research is warranted to establish the relationship between *PLEKHS1* and the IGF axis in BCa, as well as how such putative relationships overlap with T2DM and obesity and the pathogenesis and progression of BCa. This could lead to the identification of novel biomarkers or potential drug targets that could result in improvements in the diagnosis and treatment of patients. 

## Figures and Tables

**Figure 1 ijms-22-11150-f001:**
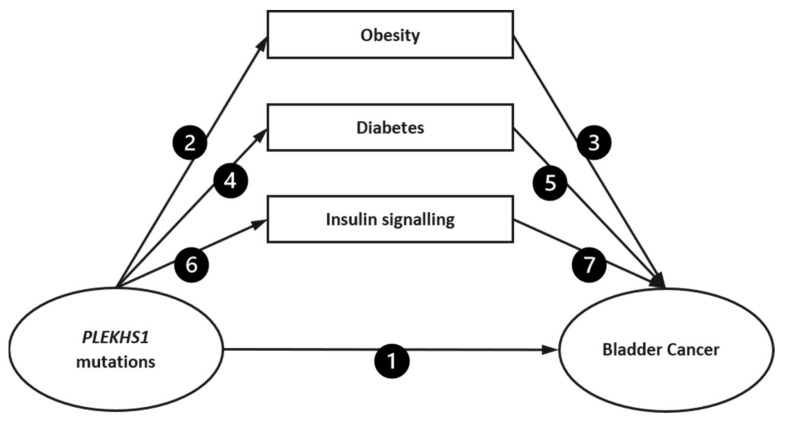
Search strategy. The figure shows the seven literature searches performed: (1) PLEKHS1 mutations and bladder cancer incidence and outcomes, (2) PLEKHS1 mutations and obesity, (3) obesity and bladder cancer incidence and outcomes, (4) PLEKHS1 mutations and diabetes, (5) diabetes and bladder cancer incidence and outcomes, (6) PLEKHS1 mutations and insulin signalling, (7) insulin signalling and bladder cancer incidence and outcomes.

**Figure 2 ijms-22-11150-f002:**
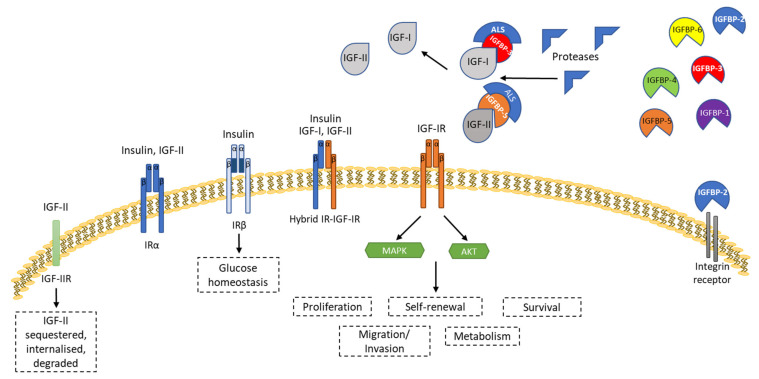
Overview of the intracellular signalling of the IGF system. At the cellular level, IGF-I, IGF-II, and insulin ligands interact with a family of signalling tyrosine kinase receptors: the IGF-IR and the insulin receptor IR, which exists in two alternatively spliced isoforms (IRα and IRβ). IRβ has a high affinity for insulin, whereas IRα has a high affinity for IGF-II. Upon the binding of the ligands to the receptors, a signalling cascade is initiated, resulting in the activation of the PI3K/Akt/mTOR/S6K and Grb2/SOS/Ras/Raf/MEK/ERK pathways. Such a cascade culminates in increased cell proliferation, survival, self-renewal, homeostasis, and metabolism. IGFs in the circulation are transported in combination with IGFBP-3 or -5 and an acid labile sub-unit (ALS) that increases their half-life. IGFs are released from IGFBPs -3 and -5 through the action of proteases. There are six high-affinity IGFBPs [1–6] that can act in either an IGF-dependent or independent manner. IGFBPs can interact with different cell surface molecules to exert their IGF-independent effects—for example, integrin receptors or ‘putative’ IGFBP receptors.

**Figure 3 ijms-22-11150-f003:**
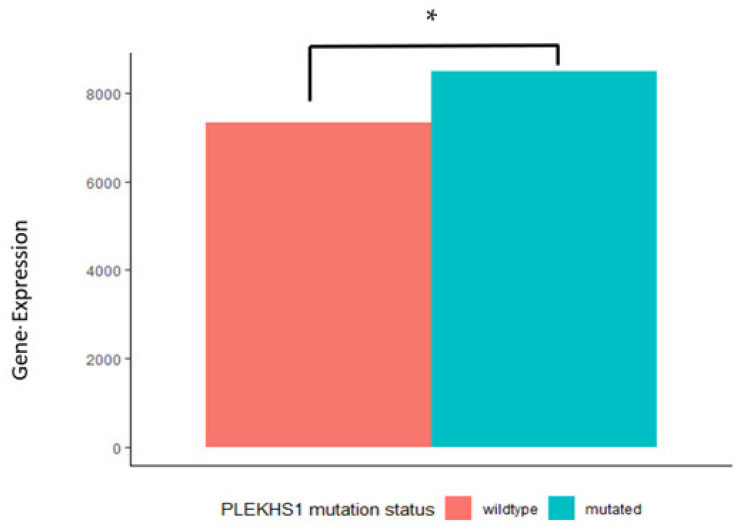
The median mRNA levels of PLEKHS1 and the IGF axis were plotted for those with and without the PLEKHS1 mutation. IGFBP3 mRNA levels were significantly increased in those carrying the PLEKHS1 mutation compared to those with the wildtype (* *p* = 0.05). Wilcoxon sum of ranks test was used.

**Figure 4 ijms-22-11150-f004:**
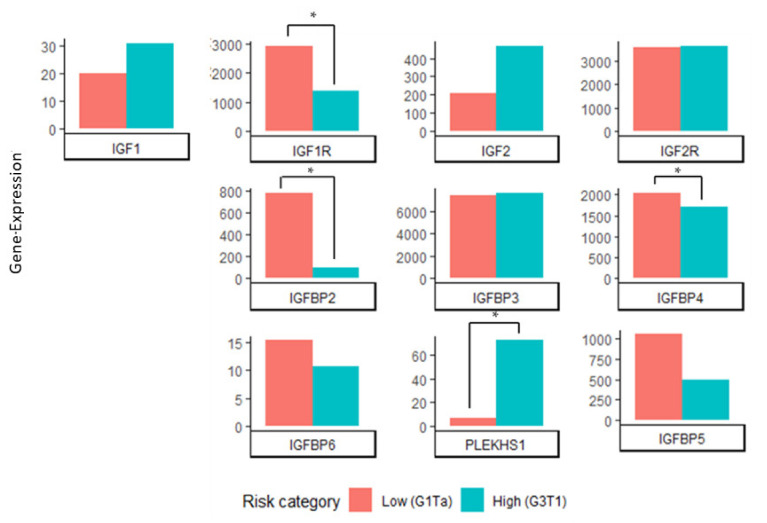
mRNA expression of the PLEKHS1 gene is significantly increased in G3T1 tumours compared to G1Ta. IGFBP2, IGFBP4, and IGF1R are significantly decreased in G3T1 compared to G1Ta tumours. Different *Y*-axes were used for each gene to improve the data visualisation. (* *p* = 0.05).

**Table 1 ijms-22-11150-t001:** Obesity and the incidence of bladder cancer.

Study Author	Study Year	Total No. of Patients	Total No. of Studies	Study Type	Comments
Qin et al. [21]	2013	8,718,502	11 (includes studies A–J)	Meta-analysis of cohort studies	“When stratifying by gender, the summary RRs with 95% CIs were 1.10 (95% CI 1.05–1.16; *p* = 0.334 for heterogeneity; I2 =12.3%) for male, and 1.15 (95% CI 1.02–1.29; *p* = 0.190 for heterogeneity; I2 =29.8%) for female”. “Among the 9 studies that controlled for cigarette smoking, the pooled RR was 1.09 (95% CI 1.01–1.17; *p* = 0.131 for heterogeneity; I2 = 35.9%)”.
Noguchi et al. [19]	2015	8,920,237	16 (includes studies A–F, and I–K)	Review	“The single largest study identified a null association of obesity with bladder cancer incidence”.
Stewart et al. [22]	2011	Not applicable	N/A	Review	“Although, a relationship between obesity and the natural course of bladder cancer may be present, due to the mixed and minimal observations within the literature, no firm conclusions can be drawn at this time”.
Zhao et al. [23]	2017	5,640,760	14 (includes studies A–K)	Meta-analysis	“There was evidence of heterogeneity among studies for obesity category (*p* = 0.003, I^2^ = 58.5%)”.
Sun et al. [18]	2015	14,201,500	15 (includes studies A–H, J, and K)	Meta-analysis of cohort studies	“Stronger associations between BMI and bladder cancer risk were found if BMI was assessed by self-reported, and if the average age of participants was greater than 50 years old. No significant effect differences were observed for duration of follow-up and for the gender of participants”.
Eggers et al. [24]	2013	Not applicable	N/A	Review	“Conflicting literature points to an unclear, but possible relation between obesity and bladder tumors”.

A = Haggstrom et al., 2011, B = Keobinck et al., 2008, C = Larsson et al., 2008, D = Semanic et al., 2006, E = Holick et al., 2006, F = Tripathi et al., 2002, G = Jee et al., 2008, H = Reeves et al., 2007, I = Cantwell et al., 2006, J = Wolk et al., 2001, and K = Rapp et al., 2005.

**Table 2 ijms-22-11150-t002:** Obesity and the prognosis of bladder cancer.

Study Author	Study Year	Total No. of Patients	Total No. of Studies	Study Type	Comments
Noguchi et al. [19]	2015	8,920,237	7	Review	“In two studies that also examined bladder cancer progression or recurrence, both (100%) noted strong associations of obesity with these outcomes”.
Westhoff et al. [25]	2018	16,198	13 (includes studies A and B)	Systematic review and meta-analysis	“No association of BMI with risk of progression was found. Results for BMI and prognosis in muscle-invasive or in all stages series were inconsistent.”
Gild et al. [26]	2017	Not applicable	N/A	Review	“With regard to the impact of obesity on survival, no final conclusion can be drawn at this time, because past publications have yielded controversial results.”
Lin et al. [27]	2018	6452	11 (includes studies A and B)	Meta-analysis	“We did not observe a difference in the rate of cancer overall survival associated with obesity. However, obese patients were prone to shorter overall survival. The summary HR and 95% CI were 1.21 (0.97–1.52), *p* = 0.679.”

A = Chromecki et al., 2012 and B = Dabi et al., 2017.

**Table 5 ijms-22-11150-t005:** Tumour cohort characteristics.

UICC Stage	No.	WHO (1973) Grade			EAU NMIBC Risk Group			Sex		Age (yrs)	Progression to MIBC		PFS (yrs)	Death		Smoking status			
	N	Grade 1	Grade 2	Grade 3	Low	Intermediate	High	Male	Female	Median	Yes	No	Median	Yes	No	Non-Smoker	Current	Ex-Smoker	Unavailable
pTa	29	17	4	8	9	8	12	24	5	71.42	11	18	3.91	6	23	2	6	18	3
pT1	49	0	1	48	0	0	49	43	6	73.82	19	30	4.47	19	30	12	7	29	1
T2+	7	0	0	7	NA	NA	NA	6	1	76.51	NA	NA	NA	6	1	1	0	6	0

## Data Availability

Sequence data were deposited at the European Genome-phenome Archive (EGA), which is hosted by the EBI and the CRG, under accession number EGAS00001004358. Further information about EGA can be found at https://ega-archive.org under “The European Genome-phenome Archive of human data consented for biomedical research” (http://www.nature.com/ng/journal/v47/n7/full/ng.3312.html).

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
