# Peer review of "The Sirenic Links between Diabetes, Obesity, and Bladder Cancer"

_ijms, 2021, doi:10.3390/ijms222011150_

Round 1
Reviewer 1 Report
The manuscript named „The sirenic links between diabetes, obesity, and bladder cancer” reviews a relevant topic – the link between bladder cancer, obesity and diabetes.
The manuscript is designed and written well.
In the abstract part, there is a lot of abbreviations that are explained, such as T2DM, IGD BCa, but PLEKHS1 not. Please, fix it.
I miss the list of abbreviations.
In figure 1, there are numbers 1-7, but they are not explained in the figure legend. Because the figure should have self-explanatory value, because it could stay alone without text, the information should stay in the legend. Please, add it.
On page 3, the first paragraph at the end “…making it candidate biomarker for bladder cancer”. Bladder cancer was previously defined through abbreviation, please use it throughout the whole manuscript.
On the page 9/15, the paragraph IGF-I and IGF-II, the name Probst-hench should be Probst-Hench. Please, correct it.
Figures 3 and 4 should contain in the legend also a citation for data used. Please, add the information.
The link in citation 52 is not correct.
Author Response
We thank the reviewer for reading our paper and for their constructive comments. All amendments have been highlighted in yellow in the manuscript together with a point-by-point rebuttal.
The manuscript named „The sirenic links between diabetes, obesity, and bladder cancer” reviews a relevant topic – the link between bladder cancer, obesity, and diabetes. The manuscript is designed and written well.
Thank you for these comments.
- In the abstract part, there is a lot of abbreviations that are explained, such as T2DM, IGF, BCa, but PLEKHS1 not. Please, fix it.
PLEKHS1 has now been written in full in the abstract.
- I miss the list of abbreviations.
We apologise for this oversight and a list of abbreviations has now been added on page 1 before keywords.
- In figure 1, there are numbers 1-7, but they are not explained in the figure legend. Because the figure should have self-explanatory value, because it could stay alone without text, the information should stay in the legend. Please, add it.
Again, we apologise for the oversight. The following legend has been added to Figure 1 on page 3: ‘Figure shows the seven literature searches performed. 1 – PLEKHS1 mutations and bladder cancer incidence and outcomes, 2 – PLEKHS1 mutations and Obesity, 3 – Obesity and bladder cancer incidence and outcomes, 4 – PLEKHS1 mutations and Diabetes, 5 – Diabetes and bladder cancer incidence and outcomes, 6 – PLEKHS1 mutations and Insulin signalling, 7 – Insulin signalling and bladder cancer incidence and outcomes.’
- On page 3, the first paragraph at the end “…making it candidate biomarker for bladder cancer”. Bladder cancer was previously defined through abbreviation, please use it throughout the whole manuscript.
We have changed bladder cancer to BCa.
- On the page 9/15, the paragraph IGF-I and IGF-II, the name Probst-hench should be Probst-H Please, correct it.
Thank you; this has been amended accordingly.
- Figures 3 and 4 should contain in the legend also a citation for data used. Please, add the information.
This information has now been added to the legends of figures 3 and 4 .
- The link in citation 52 is not correct.
Thank you for noticing that the last digit had been omitted: https://ega-archive.org/studies/EGAS00001004358. This has now been amended.
Reviewer 2 Report
- Do you have any clinical, pathological data? CIS (primary, secondary, concurrent), number of tumors, tumor size, intravesical therapy?
- What about other members of IGFBP 1, 4 and 6?
- What was control - normal tissue samples?
Author Response
We thank the reviewer for reading our paper and for their constructive comments. All amendments have been highlighted in yellow in the manuscript together with a point-by-point rebuttal.

Reviewer 3 Report
In the manuscript by Gill et al, the relationship between diabetes, obesity, and bladder cancer is discussed with a main focus on the role of PLEKHS1. The manuscript is well written and clear but I would request minor adjustments before considering it for publication.
- Please, adjust the reference style and acronyms according to the journal requirements.
- I suggest the authors highlight more PLEKHS1 (the main topic of this review) in both the title and the abstract.
- Please, revise the tables included in the manuscript:
- remove the column “No. of unisex/male/female studies, respectively”;
- results with p-value>0.05 means that there is no difference (for example, in Table 1 the study of Qin et al);
- use the same format for p-value avoiding to use p, P, p-value
- define all acronyms at the bottom of the table
- Results and comments are difficult to read. I suggest merging the columns and describing them shortly
- Few meta-analyses share the same studies. Please make it clearer in a separate figure. For example, both Qin et al and Noguchi et al cited the study of Larsson et al. in Sweden with 45,906 subjects
- The authors mention their previous study where they sequenced the RNA of bladder cancer tissue. Their study should be compared with publicly available data on bladder cancer, such as TCGA. You can freely download here http://firebrowse.org/?cohort=BLCA&download_dialog=true Interestingly, I saw only 5 patients of 395 with detected mutation of PLEKHS1. TGCA repository includes as well as RNAseq data to compare PLEKHS1 (C10orf81) with other genes
- Please revise the quality of the figures. The background is not necessary. Use the labels directly on the histograms so you do not need to use a color code
